# Analysis and Modification of Methods for Calculating Axial Load Capacity of High-Strength Steel-Reinforced Concrete Composite Columns

**DOI:** 10.3390/ma14226860

**Published:** 2021-11-14

**Authors:** Jun Wang, Yuxin Duan, Yifan Wang, Xinran Wang, Qi Liu

**Affiliations:** School of Civil Engineering, Northeast Forestry University, Harbin 150000, China; sunwangyifan@163.com (Y.W.); xinran19972021@163.com (X.W.); liu_q_wangyi@163.com (Q.L.)

**Keywords:** high-strength steel, steel-reinforced concrete column, axial bearing capacity, confinement effect, simulation analysis

## Abstract

To investigate the applicability of the methods for calculating the bearing capacity of high-strength steel-reinforced concrete (SRC) composite columns according to specifications and the effect of confinement of stirrups and steel on the bearing capacity of SRC columns. The axial compression tests were conducted on 10 high-strength SRC columns and 4 ordinary SRC columns. The influences of the steel strength grade, the steel ratio, the types of stirrups and slenderness ratio on the bearing capacity of such members were examined. The analysis results indicate that using high-strength steel and improving the steel ratio can significantly enhance the bearing capacity of the SRC columns. When the slenderness ratio increases dramatically, the bearing capacity of the SRC columns plummets. As the confinement effect of the stirrups on the concrete improves, the utilization ratio of the high-strength steel in the SRC columns increases. Furthermore, the results calculated by AISC360-19(U.S.), EN1994-1-1-2004 (Europe), and JGJ138-2016(China) are too conservative compared with test results. Finally, a modified formula for calculating the bearing capacity of the SRC columns is proposed based on the confinement effect of the stirrups and steel on concrete. The results calculated by the modified formula and the finite element modeling results based on the confinement effect agree well with the test results.

## 1. Introduction

Due to enjoying the advantages of strength, plasticity, toughness and weldability, high-strength steel has become one of the most important building materials. In recent years, high-strength steel with a yield strength ranging from 460 to 960 MPa has been used in building structures [1,2,3], such as the Bird’s Nest and Water Cube in China, and the Sony Center in Germany [4]. However, the applicability of the current design and calculation methods to high-strength steel-reinforced concrete (SRC) composite structures has become challenging since they have gradually been applied to high-rise buildings and long-span structures.

In 2014, Q460 and Q690 high-strength steel were used to replace Q345 ordinary steel in SRC composite columns in the upper part of the structure in the high-rise project of Zhengzhou Greenland Central Plaza in Henan Province, China, to enhance the bearing capacity and reduce the self-weight of the structures. Thus, it is necessary to develop the design of SRC structures. However, in Specifications for Structural Steel Buildings (AISC360-16, the USA) [5], Design of Composite Steel and Concrete Structures (EN1994-1-1:2004, Eurocode) [6] and Code for Design of Composite Structures (JGJ138-2016, China) [7], the design methods of SRC composite members are primarily based on the research on ordinary steel. Yang et al. performed model tests on Q460 high-strength SRC columns with a circular section to solve this issue and found that Q460 high-strength steel enhanced the bearing capacity of the SRC columns remarkably. Nevertheless, the calculation results by the relevant specifications were only about 60% of the test results, which led to a waste of materials [8]. Zhao et al. conducted tests on the Q460 SRC-core columns to find the reason why calculation results are too conservative [9]. It was found that the confinement effect of steel and stirrups on bearing capacity was neglected in the methods for calculating the bearing capacity of SRC composite columns in current specifications, and only the load-bearing capacity of the steel, the concrete and the longitudinal reinforcements was taken into account. Moreover, the theoretical stress–strain model of confined concrete on concrete was developed, but neither the full verification of the model nor specific calculation suggestions was proposed for the bearing capacity of SRC structures.

In recent years, there has been some research on the confinement theory of SRC members. In 1992, Mirza and Skrabek conducted tests on slender composite beam-columns with ordinary strength materials [10]. It was found that concrete in a SRC cross-section can be divided into three parts according to the confining pressure level, that is: unconfined concrete (UCC), partially confined concrete (PCC) and highly confined concrete (HCC). For PCC, the confinement effect can be considered as normal reinforced concrete, which has been extensively studied by Sheikh and Uzumeri, Mander et al. [11,12]. For HCC, Chen and Wu proposed an analytical method for predicting the axial compressive behavior of the SRC column using a cross-shaped steel section with flanges [13]. However, Wang and Su carried out tests on slender SRC columns; 270 MPa~600 MPa steel was used in the specimens. It was found that when the confinement of steel and stirrup was calculated, the bearing capacity of slender SRC columns increased by only 2% compared with only considering the confinement of stirrups [14]. Despite all this research, there are still some codes for SRC columns which do not take into account the strength of confined concrete, leading to the calculations being too conservative, such as AISC, Eurocode 4 and JGJ138-2016. Therefore, more experiments are needed to analyze the confinement of stirrups and steel on concrete, especially when high-strength steel is used in SRC columns. The codes also need more suggestions to improve the accuracy of the calculations.

In this paper, structural tests were carried out on SRC columns under an axial load to verify the applicability of the current specifications to high-strength SRC composite columns and provide a calculation method considering the confinement effect for such composite columns. Then, the finite element models were utilized to prove the accuracy of the prediction method using the theoretical stress–strain model of confined concrete for engineering applications.

## 2. Experimental Investigation

### 2.1. Test Specimens

On the basis of the theory proposed by Zhao et al. [15] and the formulas described in EN1994-1-1:2004, Eurocode, the strength of steel, the steel ratio, the stirrups and the slenderness ratio are the primary factors influencing the bearing capacity of SRC columns. Thus, 10 high-strength SRC columns and 4 ordinary SRC columns were analyzed to verify the confinement effect and determine the magnitude of the known factors. Table 1 lists the main parameters of the specimens, and Figure 1 shows the labelling rule of specimens.

Four kinds of sections were selected for this study, as shown in Figure 2. Moreover, stirrup encryption and reinforcement with carbon fiber cloth were performed within 1/6 height of both ends of the column to prevent local pressure failure at the upper and lower ends of the columns.

### 2.2. Material Properties

Steel sheets of grade Q235, Q460, Q690, with a reinforcement of HRB400, were selected to manufacture the specimens. Tensile coupon tests were carried out to obtain the material properties of steel. According to the Chinese Standard GB/T 228-2010 [16], the measured material properties of the steel presented in Table 2 are the mean values of results.

C50 concrete is used in this test. After finishing the preparation work, all the specimens were poured at one time and maintained for 28 days under the condition of a temperature no less than 5 °C. According to the standard for test methods of concrete structures (GB/T 50152-2012) [17], 9 concrete cubes (150 × 150 × 150 mm) were maintained under the same conditions as the specimens. The compressive strength test was carried out before the formal loading began, as shown in Figure 3. The results are listed in Table 3.

### 2.3. Test Setup and Procedure

A 30,000 kN servo-hydraulic testing machine was employed for loading, and the loading device is shown in Figure 4a. The accuracy of the instrument is 1/1000, which is allowed according to GB50152-2012 [17]. The specimens were subjected to multi-stage loading, and before formal loading, a 50 kN load was applied in advance to confirm whether the test apparatus functioned normally. During the formal testing, force-controlled loading was first conducted at a rate of 200 kN/min. When the load reached 60% of the estimated ultimate bearing capacity, the loading rate was reduced to 150 kN/min. When the load reached 80% of the estimated ultimate bearing capacity, the loading mode was changed from the force-controlled loading to the displacement-controlled loading, and the loading rate was set as 0.4 mm/min. After the ultimate load (ultimate bearing capacity), the testing was terminated when the load decreased to 70% of the ultimate bearing capacity. The arrangement of the measuring points is depicted in Figure 4b. The strain gauges were arranged on two adjacent sides of the steel, on the surfaces of stirrups and the longitudinal reinforcement, and four sides of concrete in the middle height of the specimens. The vertical displacement of the specimens was recorded by a displacement meter in the loading device.

### 2.4. Test Results and Analysis

#### 2.4.1. Failure Mode

The specimen was in the elastic deformation stage during the initial loading, and no cracks formed on its surface. With an increase in the load, apparent longitudinal cracks appeared in the middle of the specimen and continuously extended to both ends of the column. When the ultimate bearing capacity was reached, both the steel and longitudinal reinforcement of the specimen reached the yield strength, the length and width of the cracks increased rapidly. In addition, the cover concrete at the middle height of the specimen crushed and peeled off, and the longitudinal reinforcement bulged outward.

Figure 5 illustrates the failure modes of the specimens. As shown in Figure 5a, when the specimen with built-in Q235 steel failed, it had a relatively low degree of surface cracking. At a steel ratio of 7.17%, when the strength grade of the steel improved from Q235 to Q460 and Q690, the vertical displacement of the specimen during failure increased from 7.53 mm to 10.5 mm and 11.7 mm, respectively. Furthermore, the cracking and crushing degree of the cover concrete intensified accordingly, as illustrated in Figure 5b,c, respectively.

As presented in Figure 5d, specimen C9 with a slenderness ratio of 29 showed strength failure characteristics. However, when the slenderness ratio of the specimen increased to 40, specimen C10 displayed evident instability failure characteristics with a significant lateral displacement, as shown in Figure 5e.

Figure 6 demonstrates the crushing patterns of the concrete in high-strength SRC composite columns with different stirrup configurations. Compared with the specimens with complex stirrups, the concrete crushing depth of specimens C1, C5 and C11 with standard rectangular stirrups reached the stirrup confinement concrete during failure. On the contrary, the stirrup confinement concrete of specimens C2, C6, and C12 with complex stirrups remained almost intact during failure.

#### 2.4.2. Ultimate Bearing Capacity

Table 4 lists the ultimate bearing capacity of the specimens. Compared with the specimen with built-in Q235 steel, the ultimate bearing capacity of the specimens with built-in Q460 and Q690 steel increased by 17.1% and 35.3%, respectively, indicating a marked increase in the bearing capacity of the columns. Figure 7a plots the load–displacement curves of the specimens at different strength grades of the steel and steel ratios. For the specimens with the same strength grade of steel, the maximum improvement in the ultimate bearing capacity of the specimens with a steel ratio of 5.63% and 7.17% was 13.2% and 28.3%, respectively, compared with that of the specimen with a steel ratio of 4.12%, which implied that increasing the steel ratio could noticeably enhance the ultimate bearing capacity of the members.

Figure 7b delineates the load–displacement curves of the specimens at different slenderness ratios. Raising the slenderness ratio from 17 to 29 while keeping the other parameters unchanged reduced the ultimate bearing capacity of the column by 4.9%, implying a negligible reduction in the bearing capacity of the members. In contrast, when the slenderness ratio increased from 17 to 40, the ultimate bearing capacity of the column declined to 90.1%, indicating a remarkable reduction. Specimen C10, with a slenderness ratio of 40 experienced a small vertical displacement when the load reached the ultimate bearing capacity. Moreover, the ultimate bearing capacity of the specimen plummeted, and the specimen presented evident brittle failure characteristics.

Figure 7c draws the load–displacement curves of the specimens with different types of stirrups. When the type of stirrup was changed from standard rectangular stirrups to complex stirrups, the ultimate bearing capacity of the column with built-in Q235, Q460 and Q690 steel increased by 7.3%, 13.2% and 11.6%, respectively. The strength grade of the steel also raised the bearing capacity of the high-strength SRC composite columns with complex stirrups, which was due to the profound confinement effect of the complex stirrups, bringing the steel into full play.

## 3. Modification of Bearing Capacity Calculation

### 3.1. Current Calculation Methods

In the American National Standard Specifications for Structural Steel Buildings (AISCI360-16), the wrapped reinforced concrete part is considered to be equivalent to the steel. Thus, the formula for calculating the axial compression by utilizing the steel structure design method is defined as:(1)$$P_{n}=\left\{\begin{array}{ll} P_{n0}\left(0.658^{\frac{P_{n0}}{P_{e}}}\right) & \frac{P_{n0}}{P_{e}}\leq 2.25\\ 0.877P_{e} & \frac{P_{n0}}{P_{e}}>2.25 \end{array}\right.$$
(2)$$P_{n0}=F_{y}A_{s}+F_{ysr}A_{sr}+0.85f_{c}^{\prime }A_{c}$$
(3)$$P_{e}=\pi^{2}(EI_{\mathrm{eff}})/L_{c}^{2}$$
where $A_{s}$, $A_{sr}$ and $A_{c}$ are the cross-sectional area of the section steel, longitudinal reinforcement and concrete, respectively. $F_{y}$, $F_{ysr}$ and $f_{c}^{\prime }$ represent the compressive strength of the steel, longitudinal reinforcement and concrete, respectively; $EI_{eff}$ stands for the effective stiffness of the section; $L_{c}$ is the effective length of the member.

The Code Design of Composite Steel and Concrete Structures (EN1994-1-1:2004), defines the formula for calculating the bearing capacity of biaxially symmetric SRC columns under axial compression as:(4)$$N_{Ed}\leq \chi N_{pl,Rd}$$
(5)$$N_{pl,Rd}=A_{a}f_{yd}+0.85A_{c}f_{cd}+A_{s}f_{sd}$$
where $A_{a}$, $A_{c}$ and $A_{s}$ denote the cross-sectional area of the section steel, concrete and longitudinal reinforcement; $f_{yd}$, $f_{cd}$ and $f_{sd}$ are the compressive strength of the section steel, concrete and reinforcement, respectively; χ is the buckling reduction factor considering the relative slenderness ratio, and is expressed by Equation (7), as described in the Section 6.3 of Eurocode 3: Design of Steel Structures (EN 1993-1-1:2005),
(6)$$\chi =\frac{1}{\Phi +\sqrt{\Phi^{2}-{\bar{\lambda }}^{2}}},\;\mathrm{when}\;\chi \leq 1$$
(7)$$\Phi =0.5\left[1+\alpha (\bar{\lambda }-0.2)+{\bar{\lambda }}^{2}\right]$$
where *α* is the section type; $\bar{\lambda }$ represents the relative slenderness ratio.

The Code for Design of Composite Structures (JGJ138-2016), defines the formula for calculating the bearing capacity of axially compressed SRC columns under axial compression as:(8)$$N\leq 0.9\varphi (f_{c}A_{c}+f_{y}^{\prime }A_{s}^{\prime }+f_{a}^{\prime }A_{a}^{\prime })$$
where $A_{c}$, $A_{s}^{\prime }$ and $A_{a}^{\prime }$ indicate the cross-sectional area of the concrete, reinforcement and section steel, respectively; $f_{c}$, $f_{y}^{\prime }$ and $f_{a}^{\prime }$ are the design value of the compressive strength of concrete, reinforcement and section steel, respectively; φ is the coefficient of axial compression stability and can be determined according to the slenderness ratio presented in a specific table in the code JGJ138-2016.

### 3.2. Comparison between Test Results and Calculations

Table 4 lists the test results and the calculations of the different specifications. Figure 8 compares them at various parameters. Figure 8a,b demonstrate that as the strength grade of steel improves from Q235 to Q460 and Q690 at a constant steel ratio, the results calculated by the different codes are far smaller than the test results, indicating that the calculations are too conservative. The results calculated according to code JGJ138-2016 are the closest to the test results, whereas those calculated according to code AISC360-16 are the most conservative.

Figure 8c shows that, at a constant strength grade of the steel, the slope of the growth of the test results is similar to that of the results when the steel ratio enlarges.

As shown in Figure 8d, when the slenderness ratio increases from 17 to 40, the changing trend of the bearing capacity of the column determined by the test is similar to the one calculated by code AISC360-16: the higher the slenderness ratio is, the more profound its impact on the bearing capacity of the SRC composite columns becomes. However, the calculation results of codes JGJ138-2016 and Eurocode 4 show a small decreasing trend in the bearing capacity of the SRC columns with an increase in the slenderness ratio.

According to Figure 9, since the influence of the type of stirrups on the bearing capacity of composite columns is not considered in the methods proposed by different codes, the calculations according to different codes are generally similar. Changing the types of stirrups and strengthening the stirrup confinement effect on SRC composite columns can enhance the ultimate bearing capacity of the high-strength SRC columns more than that of the ordinary SRC columns. The ultimate bearing capacity of the Q690 SRC composite columns is relatively low when rectangular stirrups are configurated.

### 3.3. Modification of Formula for Calculating Bearing Capacity Considering Confinement Effect of Stirrup

Specifications ignore the confinement effect of the stirrups and steel on concrete when calculating the bearing capacity of reinforced concrete columns under axial compression. The bearing capacity calculated by the different specifications is smaller than the measured bearing capacity, thus they produce a conservative result. At present, there are two types of methods for analyzing the confinement effect: one only considers the confinement effect of stirrups [12], and the other takes account of the confinement effect of the stirrups and steel [15,18].

Figure 10a illustrates the section considering the confined effect of the stirrups on concrete, and Figure 10b shows the section considering the confinement effect of the stirrups and the steel on concrete. It is found that the confinement effect of the steel (open section) on concrete increases the bearing capacity of the composite columns marginally; that is, by less than about 2% [14]. Hence, this theoretical analysis only takes the confinement effect of the stirrups on the concrete strength into account. There are two methods to calculate the stress–strain relationship of stirrup confined concrete.

One is according to the research of Uzumeri and Mander [11,12], the maximum restraint stress on the stirrups only plays a role in the core confined area. Thus, Mander proposed the following calculation formula for the effective restraint stress on the stirrups:(9)$$f_{l}^{\prime }=\frac{1}{2}k_{e}\rho_{s}f_{yh}$$
where $\rho_{s}$ is the stirrup ratio, $f_{yh}$ represents the yield strength of the stirrup and $k_{e}$ indicates the effective restraint coefficient of the stirrup and is given by Equation (10). The effect of the stirrup on the core concrete confinement area is regarded as the one on all concrete areas within the stirrup area.
(10)$$k_{e}=\frac{\left(1-\sum \frac{{\left(w^{\prime }\right)}^{2}}{6b_{c}d_{c}}\right)\left(1-\frac{s^{\prime }}{2b_{c}}\right)\left(1-\frac{s^{\prime }}{2d_{c}}\right)}{1-\rho_{cc}}$$
where $\omega^{\prime }$ is the net distance between the adjacent longitudinal reinforcement; $b_{c}$ and $d_{c}$ represent the length and width of the rectangular stirrup, respectively; $s^{\prime }$ denotes the net distance between the stirrups; $\rho_{cc}$ stands for the ratio of the area of the longitudinal reinforcement to that of the confinement area.

Then, Mander developed the calculation method for the peak stress improvement coefficient k of concrete in the confinement area:(11)$$k=-1.254+2.254\sqrt{1+7.94\frac{f_{l}^{\prime }}{f_{c0}}}-2\frac{f_{l}^{\prime }}{f_{c0}}$$
where $f_{c0}$ is the axial compressive strength of concrete.

The other is the calculation method in the Section 7.2.3.1.6 of fib-CEB Model Code 2010 [19].
(12)$$\sigma_{2}=w_{c}f_{cd}(1-\frac{s_{c}}{a_{c}})(1-\frac{s_{c}}{b_{c}})(1-\frac{{\sum b}_{i}^{2}/6}{a_{c}b_{c}})$$
(13)$$k=\frac{f_{ck,c}}{f_{ck}}=1+3.5{\left(\frac{\sigma_{2}}{f_{ck}}\right)}^{\frac{3}{4}}$$
(14)$$\varepsilon_{c2,c}=\varepsilon_{c2}\left[1+5\left(\frac{f_{ck,c}}{f_{ck}}-1\right)\right]\;$$
where $\sigma_{2}\left(=\sigma_{3}\right)$ is the effective lateral compressive stress at the ULS due to confinement; $f_{ck}$ stands for the characteristic compressive strength of concrete; $f_{ck,c}$ is the value of confined concrete.

In terms of application, the above equations are very complicated, especially for square section members. To simplify, Yu Xiaolai (a scholar) proposed simplified formulas through a number of tests based on Mander’s theory [20]. The equations are as follows,
(15)$$f_{cc}=f_{c0}(2.254\sqrt{1+3.85\lambda_{v}}-0.97\lambda_{v}-1.254)\;k=\frac{f_{cc}}{f_{c0}}$$
(16)$$\varepsilon_{cc}=\varepsilon_{c0}(1+3.5\lambda_{v})$$
(17)$$\lambda_{v}=\rho_{s}f_{yh}/f_{c0}$$
where $\varepsilon_{c0}$ is the strain corresponding to the peak stress on the unconfined concrete, $\rho_{s}$ indicates the stirrup ratio; $f_{c0}$ denotes the axial compressive strength of concrete; $f_{yh}$ stands for the yield strength of the stirrup; $\lambda_{v}$ stands for the stirrup eigenvalue.

Based on the research by Yu, Equations (15)–(17) also apply to circular section members. The calculations of the simplified formulas are in good agreement with those of the above two methods, so the method for calculating the bearing capacity of the SRC column of rectangular section is modified by the simplified formulas.

These test results show that the confinement effect of the stirrup on concrete is noticeable, especially when the high-strength steel is configurated. The influence coefficient of steel strength is proposed based on Mander’s model. The modified calculation formula for the bearing capacity of SRC columns under axial compression according to code JGJ138-2016 is as follows:(18)$$N\leq 0.9\varphi (kA_{c}f_{c}+A_{a}f_{a}+A_{y}f_{y})\;$$

For the SRC columns of circular or square section, Equations (15)–(17) can be selected to calculate. When the section is rectangular, Equations (9)–(14) should be used.

Figure 11 compares the bearing capacity of the specimens measured by the test results, with that calculated by the modified formulas, Equations (15)–(18), and calculations according to code JGJ 138-2016. The results confirm that the bearing capacity of the column calculated by Equations (15)–(18) deviates from the test results by only around 10%, and the bearing capacity calculated by code JGJ138-2016 deviates from the test results by 17–35%, respectively. Thus, the modified formula considering stirrup confinement can more accurately predict the ultimate bearing capacity of such members.

According to the work of Kim [21,22], it is necessary to ensure that the steel strain corresponding to the peak stress on concrete is not smaller than the yield strain of steel so as to achieve the full mechanical performance of high-strength steel in structures, that is:(19)$$\varepsilon_{cc}\geq f_{a}/E_{a}$$
where $f_{a}$ and $E_{a}$ represent the yield stress and elastic modulus of section steel, respectively.

According to Equations (16) and (17), when the high-strength steel is applied to SRC columns with a square section, this study determines the stirrup configuration conditions to ensure the full utilization of the strength of the steel as follows:(20)$$\lambda_{v}\geq \frac{f_{a}-E_{a}\varepsilon_{c0}}{3.5E_{a}\varepsilon_{c0}}$$

According to Equation (16), the minimum stirrup eigenvalues are, respectively, 0.13 and 0.35, when Q460 and Q690 steel give full play to their strength in this test. When the rectangular stirrups are configured, the stirrup eigenvalue of specimen C5 is 0.15, which is higher than the minimum stirrup eigenvalue of the SRC columns with Q460 steel, 0.13; thus, the yield strength of Q460 steel can be brought into full play. However, for specimen C11, the stirrup eigenvalue of the rectangular stirrups is far lower than 0.35, thus the effective strength of Q690 steel is only 55.1% of its yield strength.

When the complex stirrups are configured in the SRC column, the stirrup eigenvalue of the specimen C6 is 0.34, much higher than 0.13, and the stress on the Q460 steel can reach the yield strength of the Q460 steel. Nevertheless, the stirrup eigenvalue of the specimen C12 is still slightly lower than the minimum stirrup eigenvalue, 0.35. The maximum stress on the Q690 steel can only reach as high as 89% of its yield strength, as shown in Figure 12. To summarize, the test data collected in this test, the requirements in code JGJ138-2016 and the calculations of Equation (16) can provide a reference for improving the stirrup design of high-strength SRC composite columns.

## 4. Verification of Finite Element Model and Parametric Study

### 4.1. Establishment of Finite Element Model

In Section 3.2, the reason for Q690 steel (in the specimen C_11_) unyielding under ultimate bearing capacity is well explained, and the design suggestion of the high-strength SRC column is put forward. However, it is found that not only the type of stirrups but also the spacing of stirrups affect the value of $\rho_{s}$ and $\lambda_{v}$. In this test, the value of $\lambda_{v}$ was increased by changing the stirrup type, the contribution of Q690 steel was increased, but the stirrup spacing was not considered. Therefore, the influence range of stirrup spacing on the bearing capacity of the specimens was studied by the analysis of expanding parameters with finite element models.

On the basis of the material property test, the finite element model of the test specimens was established using ABAQUS software to verify the applicability of the confinement theory to high-strength SRC columns in this test and study the influence of stirrup spacing on bearing capacity for high-strength SRC columns. Figure 13 displays the typical specimen models according to the test parameters. Moreover, eight-node hexahedral linear reduction integral stress element (C3D8R) were used for the concrete and steel in the finite element model, and two-node three-dimensional truss elements (T3D2) were employed for the reinforcement. Friction was defined at the interface between the concrete and the steel to account for their bonding. The grid was divided according to the length/Width/depth = 1.0:1.0:2.5, and the confinement effect was imposed according to the actual loading device.

Table 2 presents the material properties of the steel obtained from the coupon tests. As for the concrete, three types of confinement effects were considered. Figure 14 also presents the three types of concrete sections. Figure 14a represents the section with no confinement; Figure 14b stands the section with the confinement effect of the stirrups and Figure 14c represents the section with the confinement effect of stirrups and steel. The uniaxial compression stress–strain curve of the concrete without any confinement effect was determined using the design of Concrete Structures (GB50010-2010) [23]. The stress–strain relationship of the concrete with confinement effect of the stirrups or of both stirrups and steel was calculated by Equation (21) [12]:(21)$$\sigma =\frac{f_{cc}xr}{r-1+x^{r}}$$
(22)$$x=\varepsilon /\varepsilon_{cc}\cdot r=E_{c}/\left(E_{c}-E_{sec}\right)\cdot E_{sec}=f_{cc}/\varepsilon_{cc}\cdot f_{cc}=kf_{c0}\cdot \varepsilon_{cc}=[1+5(k-1)]\varepsilon_{c0}$$
where σ is the stress of confined concrete; $f_{cc}$ is the compressive strength of confined concrete; $E_{c},E_{sec}$ are the elasticity modulus and secant modulus of concrete, respectively; $f_{c0}$ is the axial compressive strength of the unconfined concrete; k denotes the coefficient of improvement in the strength and strain.

### 4.2. Verification of Finite Element Model

Table 5 tabulates the ultimate bearing capacity of the specimens simulated by three finite element models. When the confinement effect is not taken into account, the difference between simulation data and the test results of the specimens with built-in Q235 steel is less than 5%. Figure 15a compares the simulation results with test result of C_4_. Figure 15b shows the load–displacement of simulation results and test result of C_6_, which also demonstrates the difference of specimens with built-in Q460 is between 4% and 8%, according to Table 5. Based on Table 5 and Figure 15c, the difference between the simulation data and the test results of Q690 high-strength SRC composite columns is in the range of 9% to 13%.

In general, the difference between the simulated data and the test results are both less than 11% for the models considering only the confinement effect of the stirrups and the models considering the confinement effect of the stirrups and the steel. The simulated value of the model considering the confinement effect of both the stirrups and the steel increases by about 2% compared with that of the model only considering the confinement effect of the stirrups. Therefore, the confinement effect of the stirrups and the steel on the bearing capacity of high-strength SRC composite columns cannot be ignored.

### 4.3. Finite Element Parametric Study

In order to expand the parametric analysis, it is necessary to verify the accuracy and applicability of the finite element model to high-strength SRC columns; C14 and C6 were taken as examples to compare the failure mode determined by the numerical simulation with the tested one, as shown in Figure 16 and Figure 17. It can be seen that when the ultimate bearing capacity of the SRC column is reached, the maximum stress on the reinforcement is 460 MPa. However, the maximum stress on the Q690 steel is far from its yield strength, with slight buckling. The concrete on the buckling side crushes first. For specimen C6 with complex stirrups and Q460 steel, when the bearing capacity is reached, almost the entire length of the reinforcement yields with lateral expansion. The strain of the confinement concrete increases, and it functions well together with steel, which improves the utilization of the steel, concrete and reinforcement. As a result, the maximum stress on Q460 steel exceeds its yield strength, but no apparent buckling is noticed. According to the above comparison, both the bearing capacity and the failure mode obtained from the simulation show good agreement with the test results.

Changing the stirrup type enlarges the contribution of Q690 steel effectively. Another way to improve the stirrup confinement is to reduce the spacing of stirrups. The parameters of expanded specimens and the simulation results of bearing capacity are listed in Table 6.

For the four specimens with rectangular stirrups, the increasing magnitude of ultimate bearing capacity is similar to the increasing trend of $\lambda_{v}$ with the reduction in stirrups spacing, as shown in Figure 18a. However, although the ultimate bearing capacity of the three specimens with complex stirrups increases as spacing reduces, the increasing magnitude of ultimate bearing capacity decreases when the stirrup spacing decreases from 70 to 60 mm, as shown in Figure 18b. In general, if the value of is $\lambda_{v}$, too much larger than the minimum stirrup eigenvalues, the increasing magnitude of the bearing capacity will also decrease.

According to the stress Nephogram in Figure 19, the value of $\lambda_{v}$ of the four specimens with rectangular stirrups does not reach 0.35, and the steel still does not yield under the ultimate load. However, the increasing value of ultimate bearing capacity includes both that of confined concrete and steel. For the specimens with complex stirrups, reducing the spacing from 70 to 60 mm, the increasing value of ultimate bearing capacity is mainly from the bearing capacity of confined concrete. The contribution of Q690 steel is minimal. Therefore, when the materials have been selected, the design of high-strength SRC columns should be carried out in terms of stirrup type and stirrup spacing to ensure material utilization and save materials.

## 5. Conclusions

An experimental study of SRC columns with high-strength steel was carried out to investigate the applicability of the formula for the bearing capacity described in specifications to high-strength SRC columns. It was concluded that the results calculated by the specifications were too conservative, and a modified formula considering the confinement effect of the stirrups was proposed. Furthermore, the comparison of the test results with results calculated according to the modified formula proved that considering the confinement effect of the stirrups on concrete for calculating the bearing capacity of high-strength SRC columns was accurate and effective. Finally, the finite element models considering different confinement levels were established, and their simulation results agree well with the test results. In general, an accurate calculation and design method for practical application was provided. The main conclusions that follow from the findings of the current work are that:The bearing capacity of SRC columns can be significantly improved by high-strength steel. Compared with the bearing capacity of the Q235 SRC columns, the maximum bearing capacity of the Q460 SRC column and the Q690 SRC column increase by 13.2% and 35.3%, respectively. Further, the bearing capacity of the SRC columns was significantly improved by increasing the steel ratio.When stirrups satisfy the requirements of the stirrup eigenvalues, the utilization ratio of high-strength steel increases. The bearing capacity of high-strength SRC columns with complex stirrups significantly enlarges compared with the high-strength SRC columns with rectangular stirrups.The bearing capacity of the high-strength SRC columns declines with an increase in the slenderness ratio. The high-strength SRC composite columns with a large slenderness ratio experience greater buckling deformation than high-strength SRC columns with a conventional slenderness ratio.Comparing of the results calculated by codes AISC360-16, Eurocode 4 and JGJ138-2016 with this test results reveals that these codes are too conservative, and the calculation results of code JGJ138-2016 are closest to the test results. A modified formula for the bearing capacity of the SRC columns considering the confinement effect of the stirrups on concrete is derived based on code JGJ138-2016.The simulation results considering the confinement effect of the stirrups show great agreement with the experimental bearing capacity and failure mode of the SRC columns. The contribution of high-strength steel can be maximized, and the bearing capacity can be improved by reducing the stirrup spacing. When the stirrup eigenvalues $\lambda_{v}$ is close to the minimum stirrup eigenvalues, the increasing magnitude of bearing capacity is obvious. When the stirrups eigenvalues $\lambda_{v}$ exceed the minimum stirrup eigenvalues, the improvement of bearing capacity is not apparent.

## Figures and Tables

**Figure 1 materials-14-06860-f001:**
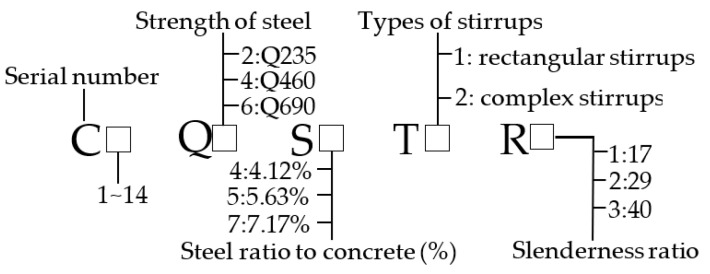
Labeling rule of specimens. (**C**: Serial number; **Q**: Strength of steel; **S**: Steel ratio to concrete; **T**: Type of stirrups; **R**: Slenderness ratio).

**Figure 2 materials-14-06860-f002:**
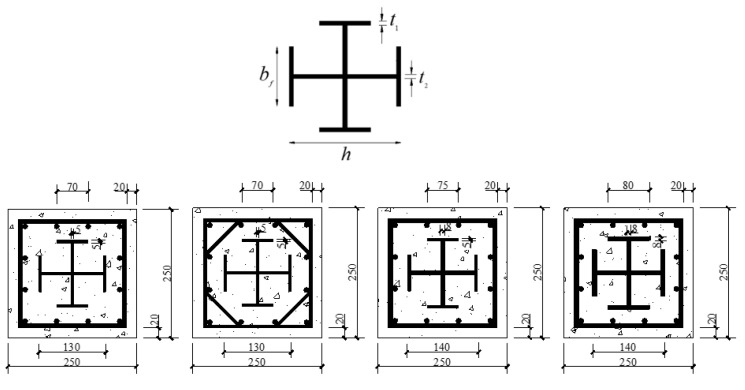
Sectional dimensions of specimens (millimeters).

**Figure 3 materials-14-06860-f003:**
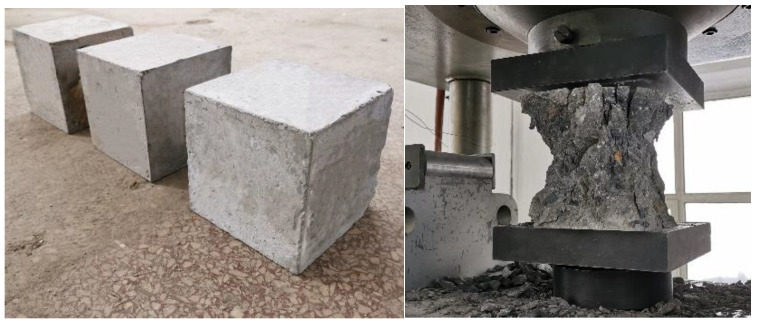
Concrete cubes and the failure mode of concrete tubes.

**Figure 4 materials-14-06860-f004:**
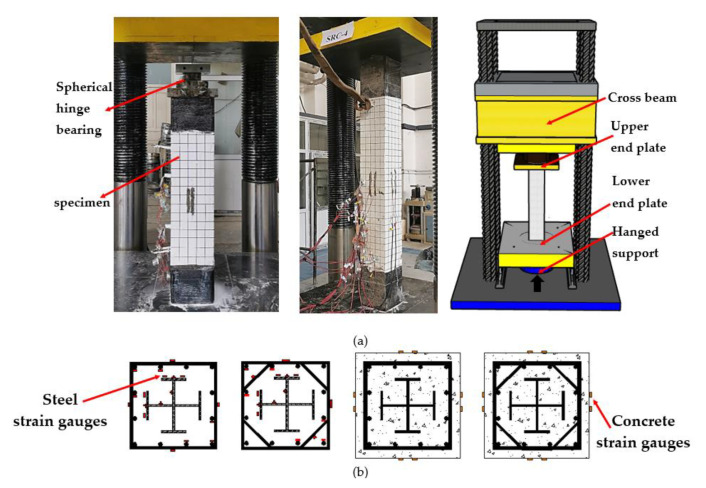
Test setup and arrangement of strain gauges. (**a**) test setup and instrumentations; (**b**) strain gauges position.

**Figure 5 materials-14-06860-f005:**
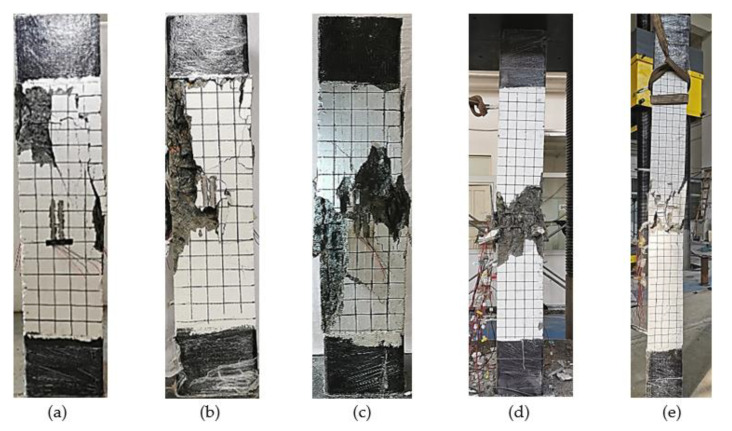
Failure modes of the specimens: (**a**) failure mode of the specimen with Q235 steel, steel ratio is 7.17%; (**b**) failure mode of the specimen with Q460 steel, steel ratio is 7.17%; (**c**) failure mode of the specimen with Q690 steel, steel ratio is 7.17%; (**d**) failure mode of the specimen C9; (**e**) failure mode of the specimen C10.

**Figure 6 materials-14-06860-f006:**
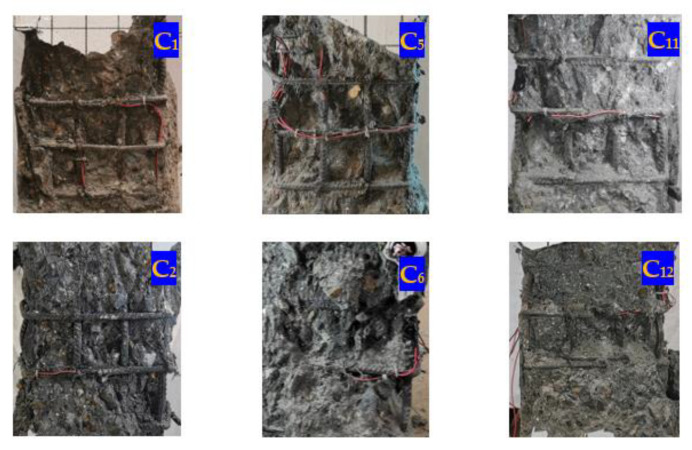
Concrete crushing modes of specimens with different stirrup configurations. (C_1_, C_5_, C_11_ are the specimens with rectangular stirrup; C_2_, C_6_, C_12_ are the specimens with complex stirrup).

**Figure 7 materials-14-06860-f007:**
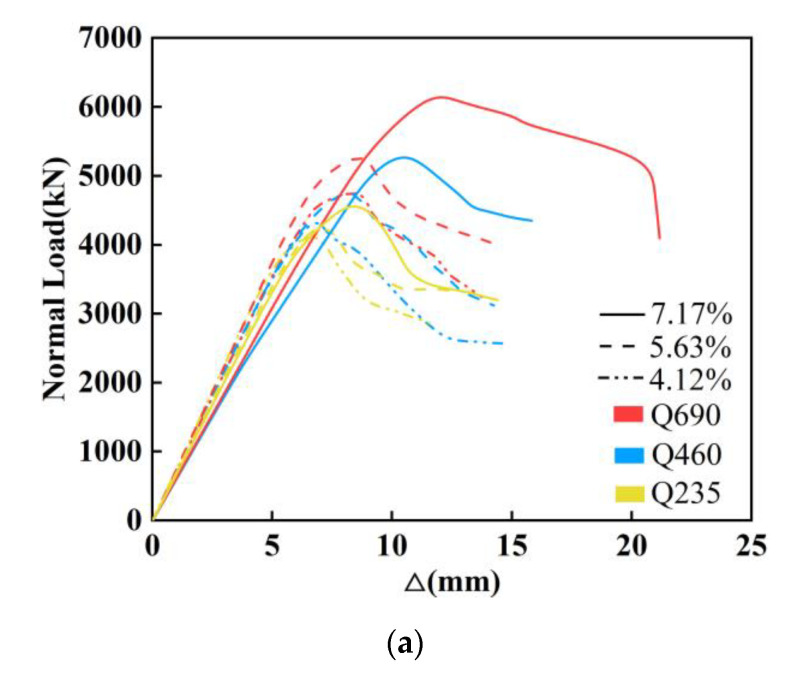
Load–displacement curves of specimens: (**a**) the load–displacement curves of the specimens at different strength grades of the steel and steel ratios; (**b**) the load–displacement curves of the specimens at different slenderness ratios; (**c**) the load–displacement curves of the specimens with different types of stirrups.

**Figure 8 materials-14-06860-f008:**
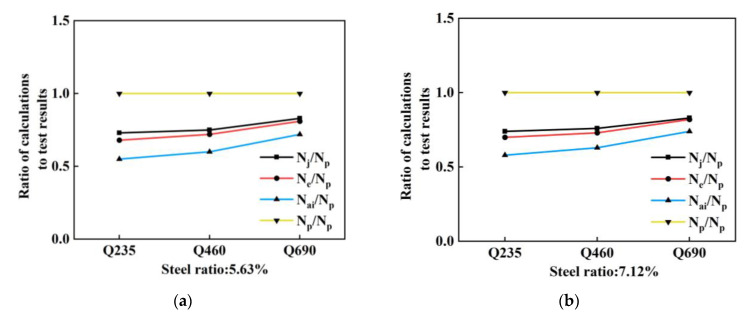
Effect of parameters on bearing capacity of specimens: (**a**) Comparison of test results and calculations for the specimens with steel ratio 5.63%; (**b**) Comparison of test results and calculations for the specimens with steel ratio 7.17%; (**c**) Comparison of test results and calculations for the specimens with Q690 steel; (**d**) Comparison of test results and calculations for the specimens with different slenderness ratios.

**Figure 9 materials-14-06860-f009:**
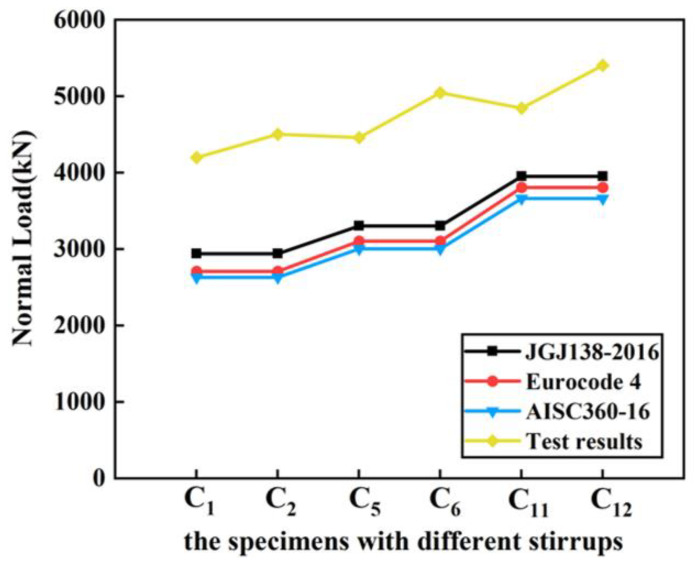
Effect of types of stirrups on bearing capacity of specimens. (C_1_, C_5_, C_11_ are the specimens with rectangular stirrup; C_2_, C_6_, C_12_ are the specimens with complex stirrup).

**Figure 10 materials-14-06860-f010:**
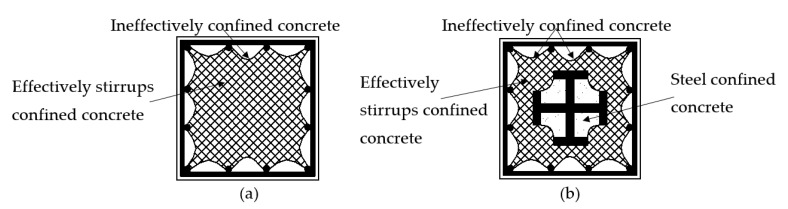
Effectively confined region and ineffectively confined region of concrete: (**a**) Ordinary reinforced concrete column; (**b**) SRC column with cross-section steel.

**Figure 11 materials-14-06860-f011:**
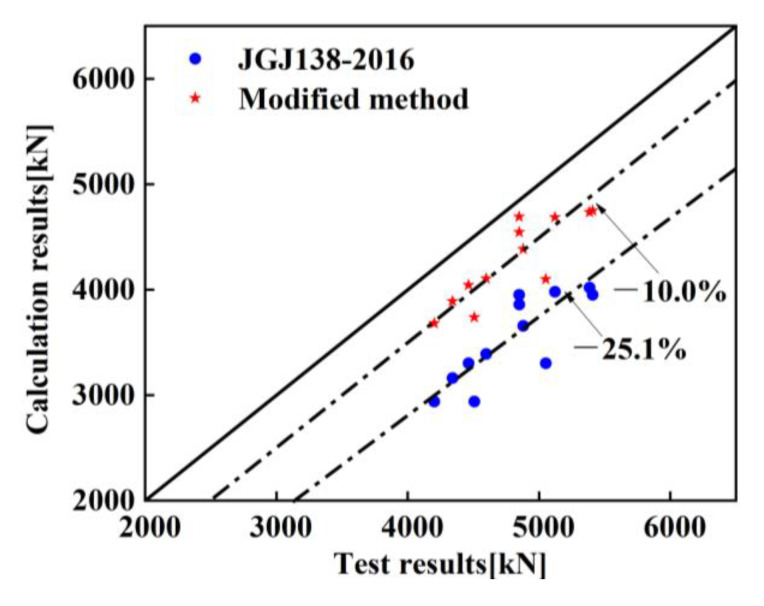
Comparison among test results, modified calculation results of formulas and the calculation results by JGJ138-2016.

**Figure 12 materials-14-06860-f012:**
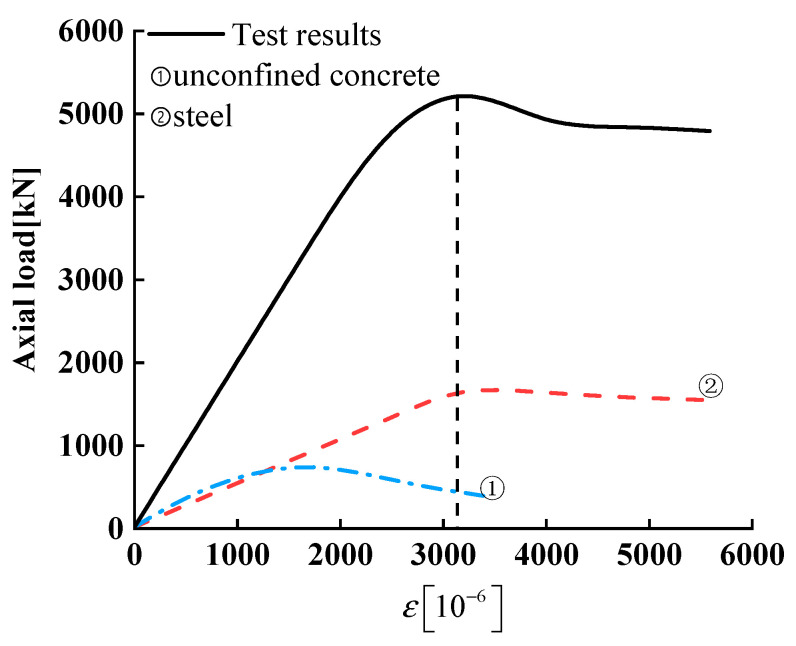
Axial load-strain relationships of C14.

**Figure 13 materials-14-06860-f013:**
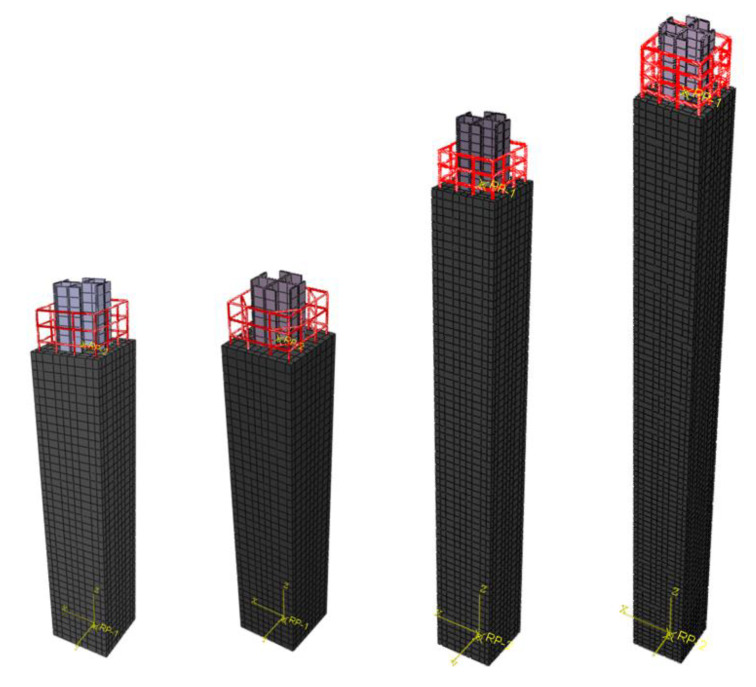
Typical specimen models.

**Figure 14 materials-14-06860-f014:**
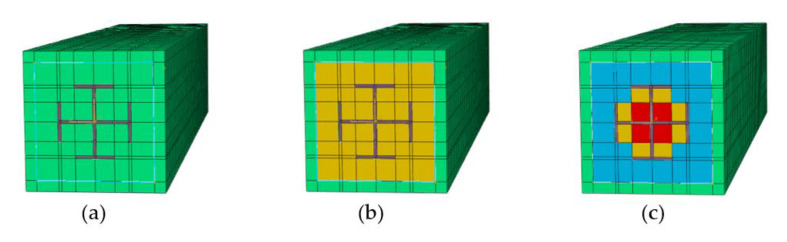
Three kinds of confinement effect on concrete: (**a**) the section with no confinement; (**b**) the section with the confinement effect of the stirrups (**c**) the section with the confinement effect of stirrups and steel.

**Figure 15 materials-14-06860-f015:**
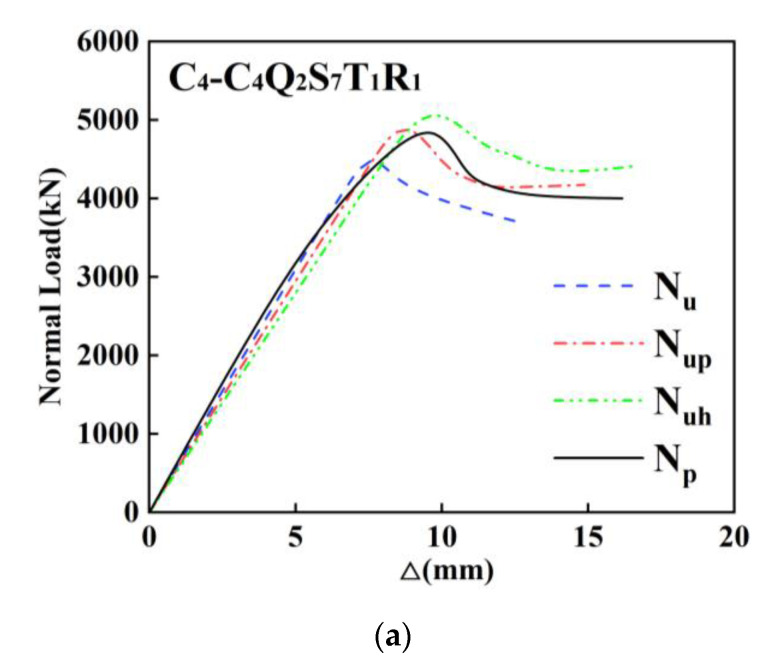
Comparison between test results and the simulation results: (**a**) the load-displacement curves of simulation results and test result for C_4_; (**b**) the load-displacement curves of simulation results and test result for C_6_. (**c**) the load-displacement curves of simulation results and test result for C_14_.

**Figure 16 materials-14-06860-f016:**
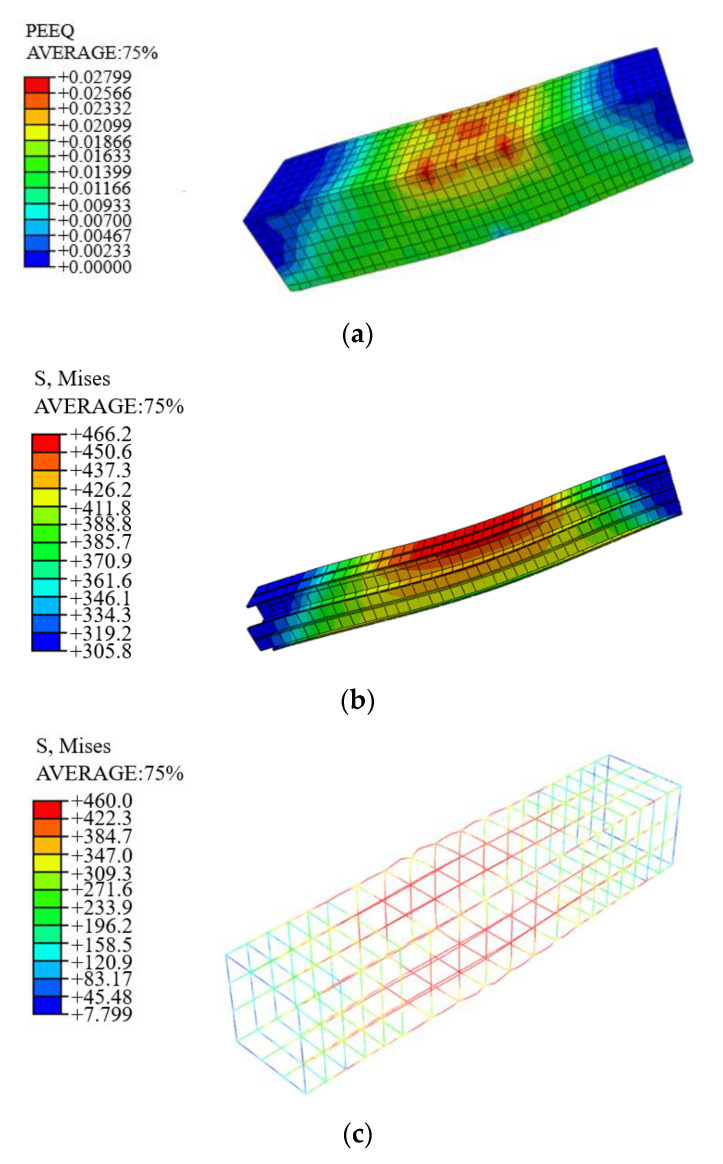
Stress–field nephogram of specimen C14 (considering only confinement effect of the stirrups). (**a**) Stress–field nephogram of C14, (**b**) Stress–field nephogram of Q690 steel in specimen C14 and (**c**) Stress–field nephogram of reinforcement.

**Figure 17 materials-14-06860-f017:**
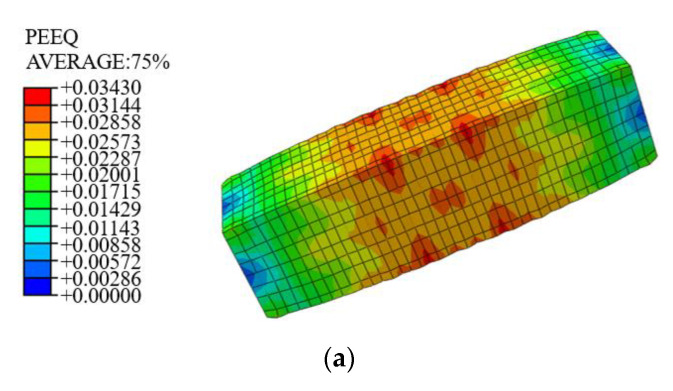
Stress–field nephogram of specimen C6 (considering only confinement effect of the stirrups). (**a**) Stress–field nephogram of C6, (**b**) Stress–field nephogram of Q690 steel in specimen C6 and (**c**) Stress–field nephogram of reinforcement.

**Figure 18 materials-14-06860-f018:**
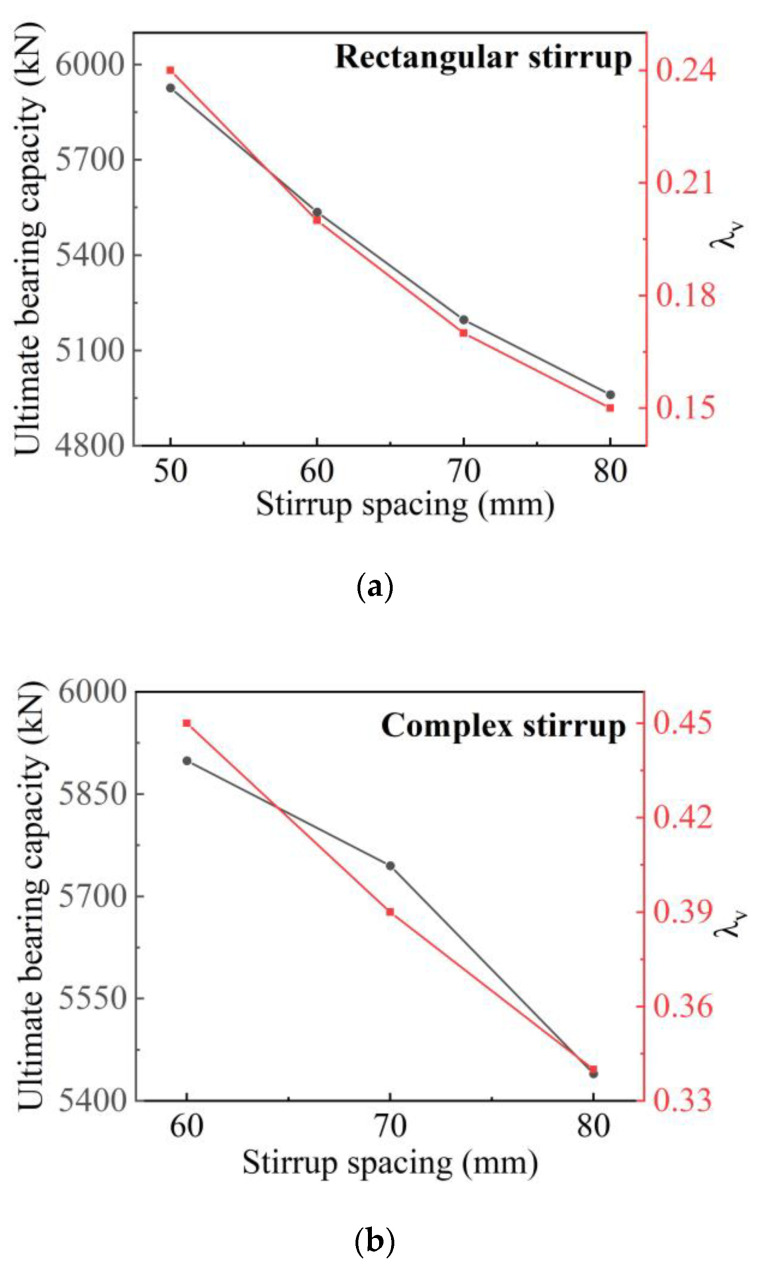
Influence of stirrup spacing on ultimate bearing capacity of the specimens: (**a**) specimens with rectangular stirrup (**b**) specimens with complex stirrup.

**Figure 19 materials-14-06860-f019:**
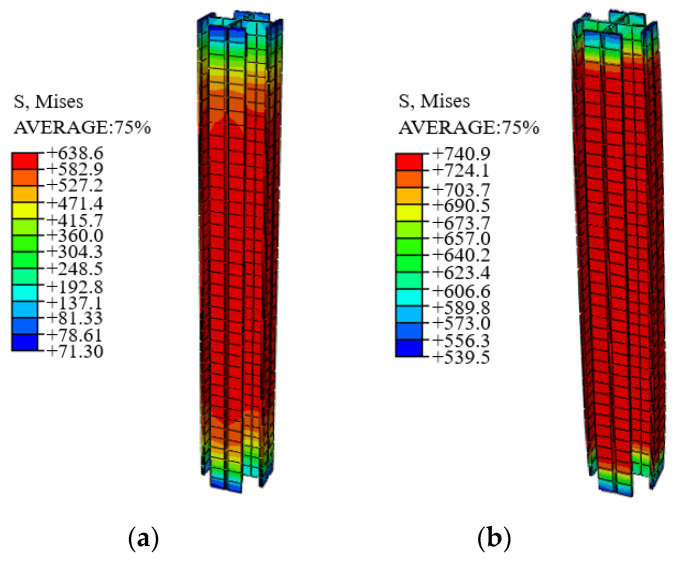
Q690 steel contribution of specimens with different stirrup spacing, type. (**a**) Rectangular stirrup, spacing = 50 mm (**b**) Complex stirrup, spacing = 70 mm.

**Table 1 materials-14-06860-t001:** Main parameters of specimens.

Specimen Designation	Steel Grade	Steel Ratio to Concrete	Steel Geometric Size (h × bf × t1 × t2)/mm	Stirrup Type	Slenderness Ratio	Height/mm	Stirrups Spacing
C1Q2S4T1R1 (C_1_)	Q235	4.12%	130 × 70 × 5 × 5	1	17	1200	80
C2Q2S4T2R1 (C_2_)	Q235	4.12%	130 × 70 × 5 × 5	2	17	1200	80
C3Q2S5T1R1 (C_3_)	Q235	5.63%	140 × 75 × 5 × 8	1	17	1200	80
C4Q2S7T1R1 (C_4_)	Q235	7.17%	140 × 80 × 8 × 8	2	17	1200	80
C5Q4S4T1R1 (C_5_)	Q460	4.12%	130 × 70 × 5 × 5	1	17	1200	80
C6Q4S4T2R1 (C_6_)	Q460	4.12%	130 × 70 × 5 × 5	2	17	1200	80
C7Q4S5T1R1 (C_7_)	Q460	5.63%	140 × 75 × 5 × 8	1	17	1200	80
C8Q4S7T1R1 (C_8_)	Q460	7.17%	140 × 80 × 8 × 8	2	17	1200	80
C9Q4S7T1R2 (C_9_)	Q460	7.17%	140 × 80 × 8 × 8	1	29	2000	80
C10Q4S7T1R3 (C_10_)	Q460	7.17%	140 × 80 × 8 × 8	2	40	2800	80
C11Q6S4T1R1 (C_11_)	Q690	4.12%	130 × 70 × 5 × 5	1	17	1200	80
C12Q6S4T2R1 (C_12_)	Q690	4.12%	130 × 70 × 5 × 5	2	17	1200	80
C13Q6S5T1R1 (C_13_)	Q690	5.63%	140 × 75 × 5 × 8	1	17	1200	80
C14Q6S7T1R1 (C_14_)	Q690	7.17%	140 × 80 × 8 × 8	2	17	1200	80

**Table 2 materials-14-06860-t002:** Mechanical properties of steel.

Grade	t/mm	fy/Mpa	fu/Mpa	δ/%
Q235	5	277	437	33.7
Q235	8	305	469	32.5
Q460	5	474	558	28.8
Q460	8	507	596	27.9
Q690	5	740	820	16.5
Q690	8	738	818	17.2
HRB400 (stirrups)	8	469	611	28.6
HRB400 (longitudinal reinforcement)	10	460	609	27.8

**Table 3 materials-14-06860-t003:** Results of compressive strength test.

Grade of Concrete	$f_{cu}^{0}$	$f_{cu,m}^{0}$	$f_{c}^{0}$	$E_{c}^{0}$
C50	52.6	55.4	40.8	35,736.3
	56.1			
	54.8			
	53.2			
	56.6			
	57.4			
	57.3			
	54.9			
	55.7			

$f_{cu}^{0}$ is the cubic compressive strength of concrete by the test, respectively; $f_{cu,m}^{0}$ is the average cubic compressive strength; $f_{c}^{0}$ is the calculation of the prism compressive strength, which is used in the finite element models; $E_{c}^{0}$ is the elastic modulus measured by the test, respectively.

**Table 4 materials-14-06860-t004:** Comparison of test results with calculation results calculated codes.

Specimens	Test Results	AISC360-16	($\frac{N_{\mathrm{ai}}}{N_{u}}$ − 1)%	Eurocode 4	($\frac{N_{e}}{N_{u}}$ − 1)%	JGJ138-2016	($\frac{N_{j}}{N_{u}}$ − 1)%
	N_u_/kN	N_ai_/kN		N_e_/kN		N_j_/kN	
C1Q2S4T1R1	4200	2633	37.3%	2713	35.4%	2943	29.9%
C2Q2S4T2R1	4506	2633	41.6%	2713	39.8%	2943	34.7%
C3Q2S5T1R1	4340	2887	33.5%	2968	31.6%	3166	27.1%
C4Q2S7T1R1	4596	3149	31.5%	3230	29.7%	3394	26.2%
C5Q4S4T1R1	4462	3006	32.6%	3107	30.4%	3305	25.9%
C6Q4S4T2R1	5050	3006	40.5%	3107	38.5%	3305	34.6%
C7Q4S5T1R1	4878	3398	30.3%	3505	28.1%	3660	25.0%
C8Q4S7T1R1	5383	3803	29.4%	3912	27.3%	4023	25.3%
C9Q4S7T1R2	5120	3561	30.4%	3778	*26.2%*	3983	22.2%
C10Q4S7T1R3	4848	3228	33.4%	3617	*25.5%*	3862	20.3%
C11Q6S4T1R1	4847	3666	24.4%	3810	21.4%	3954	18.4%
C12Q6S4T2R1	5407	3666	32.2%	3810	29.5%	3954	26.9%
C13Q6S5T1R1	5487	4299	21.7%	4461	18.7%	4546	17.1%
C14Q6S7T1R1	6220	4953	20.4%	5126	17.6%	5152	17.2%

**Table 5 materials-14-06860-t005:** Comparison of three simulation results with test results.

Specimens	ABAQUS Results			Test Results N_p_/kN	N_u_/N_p_	N_up_/N_p_	$N_{uh}/N_{p}$
	$N_{u}/\mathrm{kN}$	$N_{\mathrm{up}}/\mathrm{kN}$	$N_{\mathrm{uh}}/\mathrm{kN}$				
C1Q2S4T1R1	4107	4183	4220	4200	97.8%	99.6%	100.5%
C2Q2S4T2R1	4341	4495	4554	4506	96.3%	99.8%	101.1%
C3Q2S5T1R1	4208	4317	4388	4340	96.9%	99.5%	101.1%
C4Q2S7T1R1	4529	4690	4721	4596	98.5%	102.0%	102.7%
C5Q4S4T1R1	4270	4641	4726	4462	95.7%	104.0%	105.9%
C6Q4S4T2R1	4649	5094	5307	5050	92.1%	100.9%	105.1%
C7Q4S5T1R1	4621	4991	5056	4878	94.7%	102.3%	103.6%
C8Q4S7T1R1	5086	5447	5531	5383	94.5%	101.2%	102.7%
C9Q4S7T1R2	4824	5283	5312	5120	94.2%	103.2%	103.8%
C10Q4S7T1R3	4707	4951	5003	4848	97.1%	102.1%	103.2%
C11Q6S4T1R1	4433	4961	5019	4847	91.5%	102.4%	103.5%
C12Q6S4T2R1	4786	5440	5536	5407	88.5%	100.6%	102.4%
C13Q6S5T1R1	5025	5729	5955	5487	91.6%	104.4%	108.5%
C14Q6S7T1R1	5682	6733	6892	6220	91.1%	108.2%	110.8%

$N_{u}$ is the simulation result based on no confinement, $N_{up}$ is the simulation result only considering the confinement effect of the stirrups, $N_{uh}$ is the simulation result considering the confinement effect of the stirrups and steel.

**Table 6 materials-14-06860-t006:** Main parameters of the expanded specimens.

Specimen	Stirrup Spacing (mm)	Stirrup Type	Simulation Results (kN)	Increasing Magnitude
C11Q6S4T1R1	80	rectangular	4961	0.0%
Sp_70_-Q6S4T1R1	70	rectangular	5197	4.8%
Sp_60_-Q6S4T1R1	60	rectangular	5536	11.6%
Sp_50_-Q6S4T1R1	50	rectangular	5927	19.5%
C12Q6S4T2R1	80	complex	5440	0.0%
Sp_70_-Q6S4T2R1	70	complex	5745	5.6%
Sp_60_-Q6S4T2R1	60	complex	5899	8.4%

## Data Availability

The data presented in this study are available on request from the corresponding author.

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
