# Peer review of "Analysis and Modification of Methods for Calculating Axial Load Capacity of High-Strength Steel-Reinforced Concrete Composite Columns"

_materials, 2021, doi:10.3390/ma14226860_

Round 1
Reviewer 1 Report
It is necessary to describe in more detail the equipment and method of testing in the laboratory, some corrections and additions are needed, which are listed in more detail in the attached document with comments. The effort of the author is appreciated, with some corrections, the work will be ready for publication.

Author Response
Dear reviewer,
I'm gradeful to your advices. Thank you for your endorsement. That's all my reply, please check it.

Reviewer 2 Report
The paper is valuable, but there are a few major concerns:
- The authors say”
|
Hence, there have been few specific design and calculation suggestions considering the |
56 |
|
confinement effect on the bearing capacity of high-strength SRC composite columns un- |
57 |
|
der an axial load so far. |
“
- Please mention which attempts they are referring to, the references are inadequate!
- Please provide a figure/picture of the two adopted stirrups.
- Provide the mechanical properties of concrete.
- The confinement leads to an increment of the concrete resistance. According to the Mohr-Coulomb strength condition, the resistance of the confined concrete core can be set proportional to the confinement coefficient. This coefficient is very variable (4-7 according to the experience of the reviewer and other scholars for concrete filled steel tubes). In these structural elements, the confinement is lower than a concrete-filled steel tube. It would be interesting to calibrate the confinement coefficient, according to the Mohr-Coulomb strength condition. The authors mention a confinement coefficient in Eq.10. However, the reviewer does not see a comparison between the predicted and estimated values of the confinement coefficient. The reviewer warmly suggests including a direct estimate of the confinement coefficient using the Mohr-Coulomb approximation of the concrete resistance in a pluri-axial stress state.
- The underestimation of the ultimate capacity according to the standards depends on the neglection of the incremental confinement coefficient of concrete. The ultimate capacity is not merely the sum of the concrete core resistance and that of steel. Please indicate which are the estimates of these incremental coefficients.
-What the FE model is used for? It is not very clear in the paper. The reviewer believes that the model should be used not only for data validation but also for a possible extrapolation of the proposed formulation. Otherwise, what does the FE model add to the experimental data?
The reviewer recommends the paper for publication after addressing the above issues.
Author Response
Dear reviewer,
Thanks for your endorsement and suggestions, I really learn a lot. My reply are all in the word file, please check it.

Reviewer 3 Report
The authors present an interesting manuscript, which describes a wide-scope research including experimental tests, analytical interpretation of the results and a numerical modeling. The manuscript seems adequate for the journal and the special issue, after some revision.
The article is difficult to read in some parts due to the complex notation that has been used. For instance, the horizontal axis of Fig. 8 does not have any labelling and the meaning of C1, C2, C5, etc. is not clear. Although that information was given earlier in the manuscript, re-labelling the axes or repeating the meaning of the notation would help readers understand the points made by authors. Similarly in Fig. 6, 8 and Fig. 14, the vertical axis label is “N (kN)”, but should have been something like “Normal force”, “Normal load”, “Ultimate load”, etc. This also happens in other parts of the manuscript. For instance, the vertical axes’ labels in Fig. 7 are missing. What magnitude is represented there?
Pertaining to the analytical interpretation of the experimental results, international standards have been considered in section 3.1. This section should be extended, including the analytical models for confined concrete that might be available in those standards. For example, Eurocodes do have a model for confined concrete (cf. clause 3.1.9 in Eurocode 2, Part 1-1). In order to calculate the stress σ2 in the analytical model, basic fundamentals of strength of materials and mechanics can be used. However, in this case it could arguably be possible to find more simple and compact equations in the commentary included in the fib-CEB Model Code 2010, section 7.2.3.1.6. dedicated to confined concrete.
When trying to reproduce the analytical results, sections 2.1 and 2.2 do not describe the reinforcing steel layout, neither the longitudinal rebars, nor the stirrups’ size and spacing. This must be corrected.
Tables 1 and 3 must be revised. The complex notation used by the authors includes errors. For example, although Q690 steels have been used, those tables do not have any Q6 specimens. The ratio of structural steel area to gross concrete cross-section has different values in some parts of the document (e.g., 7.17% or 7.12%?). The manuscript includes two sections numbered as 3.2 (both on page 10).
In the presentation of the analytical results, this reviewer thinks that any reader of the manuscript could have serious doubts about the safety factors. If those analytical models in AISC, Eurocodes 2, 3 & 4 and JGJ are followed as written, safety factors for the strength of concrete and reinforcing steel are normally used –and even coefficients accounting for the strength loss due to long-term loading, etc. Hence the analytical results are surely going to be smaller than the experimental results. I have tried to work out the analytically predicted strength by one of the specimens and have used the said safety factors, reaching a value close to the predictions in Table 3 and Fig. 7. That would explain the differences without the need of creating a new model, as presented in section 3.2 “Modification of formula for calculating bearing capacity considering confinement effect of stirrups”. I must recognize that my calculations are lacking the contribution of stirrups and longitudinal reinforcement (because of the aforementioned issue with the insufficient information given in sections 2.1 and 2.2). I do not think that the contribution of longitudinal reinforcement is significant, but I was not able to estimate the enhanced compressive strength of the concrete due to the stirrup confinement (with EC2+Model Code). In any case, I recommend that authors (i) expand the information in this section; (ii) clarify whether they are using characteristic values of materials’ properties or tested values; (iii) make it clear if they are using design values of the materials’ strength, i.e., characteristic (or tested) values divided by a safety factor; and (iv) present and use other analytical procedures, different from Mander’s, to account for the increase in strength by confined concrete, when available in the international standards (such as Eurocodes).
Author Response
Dear reviewer,
I'm really gradeful to your advices. Some of your ideas are really great and helpe me a lot. My reply are all in the word file, please check it.

Round 2
Reviewer 2 Report
The paper can be accepted for publication.
Reviewer 3 Report
All my comments in the first report have been adequately addressed. I have no further comments.